# Analysis of the Prevalence of Canine Splenic Mass Lesions in Republic of Korea via Histopathological Diagnosis with Immunohistochemistry

**DOI:** 10.3390/vetsci10040247

**Published:** 2023-03-25

**Authors:** Yeong-Ung Ko, Min-Kyung Bae, Jung-Hyang Sur, Nong-Hoon Choe

**Affiliations:** 1Department of Veterinary Pathology, College of Veterinary Medicine, Konkuk University, Seoul 05029, Republic of Korea; 2Neodin Biovet Laboratory, Achiul-gil, Guri-si 11956, Republic of Korea; 3Department of Veterinary Public Health, College of Veterinary Medicine, Konkuk University, Seoul 05029, Republic of Korea

**Keywords:** dog, spleen, tumor, histopathology, immunohistochemistry, Republic of Korea

## Abstract

**Simple Summary:**

The histopathological diagnosis of canine splenic mass lesions is crucial for determining prognosis. Herein, we analyzed the prevalence of canine splenic mass lesions in Republic of Korea via histopathological diagnosis. Immunohistochemistry was performed for specific tumor markers for a more accurate diagnosis. The results of this study will aid veterinary clinicians in communication with pet owners about prognoses and recommendations for splenectomy. This study will facilitate additional investigations with more detailed comparisons of splenic mass lesions between small- and large-breed dogs.

**Abstract:**

The histopathological diagnosis of canine splenic mass lesions is crucial for prognostication. However, thus far, no study has been conducted on the histopathology of canine splenic mass lesions in Republic of Korea. Herein, the prevalence of splenic diseases was analyzed in 137 canine splenic mass lesions via histopathological diagnosis, and the microscopic pattern associated with each disorder was described. Immunohistochemistry was performed for CD31, CD3, PAX5, Iba1, and C-kit for a more accurate diagnosis of splenic tumors. The proportion of non-neoplastic disorders, including nodular hyperplasia (48.2%, n = 66) and hematoma (24.1%, n = 33), was 72.3%. Splenic tumors, including splenic hemangiosarcoma (10.2%, n = 14), splenic lymphoma (nodular and diffuse types, 8.0%, n = 11), splenic stromal sarcoma (7.3%, n = 10), myelolipoma (1.5%, n = 2), and mast cell tumors (0.7%, n = 1), accounted for 27.7% of cases. The results of this study will aid veterinary clinicians in communication with pet owners about prognoses, recommendations for splenectomy, and subsequent histopathological diagnoses. This study will facilitate further investigations with more detailed comparisons of splenic mass lesions between small- and large-breed dogs.

## 1. Introduction

Disorders of the canine spleen are increasingly being recognized with the use of advanced diagnostic equipment and higher frequencies of splenic mass lesions in aged dogs. Clinical veterinary surgeons perform splenectomies in dogs for therapeutic purposes; however, hematopoietic and immune functions are inevitably reduced in dogs that have undergone splenectomies [1]. Therefore, a decision regarding a splenectomy must be made based on an accurate and reliable diagnosis of splenic mass lesions in a dog [2].

For decades, canine splenic mass lesions have been diagnosed using histopathology for etiologic differentiation, as the tissue structure is maintained during histopathologic processing [3,4,5]. Histologic diagnosis is predominantly performed with hematoxylin and eosin (H&E) staining for the identification and evaluation of the cell types that comprise canine splenic mass lesions. If necessary, ancillary tests can be performed (e.g., immunohistochemistry) to obtain a more accurate diagnosis for canine splenic mass lesions [5]. Canine splenic mass lesions encompass hyperplastic and neoplastic (benign and malignant) processes.

Splenic neoplasia is commonly observed in older dogs and can be biologically aggressive [6]. Most malignant splenic tumors in dogs grow rapidly, and the prognosis is poor because of the risk of rupture and metastasis [6,7]. The 1-year survival rate of dogs with canine splenic lymphoma is 59.8% [8], and the mean survival time (MST) of dogs with canine splenic hemangiosarcoma is <4 months [7,9,10,11,12,13].

Splenic tumors can have various origins, including vascular, lymphoid, fibrous, smooth muscle, myeloid, and adipose tissues. Many studies have shown that the most common canine splenic tumor is splenic hemangiosarcoma [7,9,10,11,12,13], which originates from vascular endothelial cells and has a poor prognosis. There is the “double two-thirds” rule for canine splenic mass lesions, which states that two thirds of canine splenic mass lesions are malignant and that, of those, two thirds of the malignancies are hemangiosarcoma [14].

Non-neoplastic nodular diseases of the canine spleen include nodular hyperplasia and splenic hematoma. Nodular hyperplasia is a benign proliferation of cells that normally exist in the spleen and is one of the most common causes of canine splenic mass formation [11]. Nodular hyperplasia is subdivided into lymphoid, splenic, hematopoietic, and complex types, and the most common type is lymphoid nodular hyperplasia, which is characterized by the proliferation of lymphoid follicles. Each type of nodular hyperplasia is not associated with biological features, including clinical symptoms, but is classified based on the predominant cellular constituent.

Splenic hematoma is a pooling of blood in the splenic parenchyma caused by a hemorrhage or disrupted blood flow in the spleen [15]. The causes of parenchymal disruption vary, including trauma, benign nodular hyperplasia, and hemangiosarcoma, but most of them are idiopathic [15]. As the splenic capsule distends and becomes fragile, it can rupture and subsequently form a hemoabdomen.

Many studies have been conducted on the analysis of splenic mass lesions in dogs [4,11,16,17,18]. These studies were mostly carried out in Western countries, such as the United States and countries in Europe. Recently, the clinical and imaging characteristics of canine splenic mass lesions were evaluated in Republic of Korea [6]. However, the histopathology of canine splenic mass lesions has not been investigated; therefore, our aim was to provide the first histopathological analysis of canine splenic mass lesions in Republic of Korea.

Herein, we analyzed the prevalences of different canine splenic mass lesions using histopathological diagnosis and described the microscopic patterns of various splenic mass lesions. In addition, immunohistochemistry was performed for specific tumor markers for a more accurate diagnosis of splenic tumors.

## 2. Materials and Methods

### 2.1. Ethical Statement

All samples were collected from local animal hospitals via partial or total splenectomies for treatment and subsequent histopathological diagnosis. No live animals were used in this study; therefore, approval from the Institutional Animal Care and Use Committee (IACUC) was waived. Informed consent for the use of clinical data and samples for research was obtained from the owners of the dogs.

### 2.2. Sample Collection

We randomly selected 140 samples of canine splenic tissue from the histopathological database of the Department of Veterinary Pathology, Konkuk University, Seoul, Republic of Korea, between January and October 2022. Patient information, such as age, breed, and sex, was collected from clinical records. Splenic tissue samples were fixed in 10% neutral buffered formalin, processed routinely, and embedded in paraffin wax. Tissue blocks were sectioned to a thickness of 4 µm and stained with H&E. After an initial screening, three samples were excluded because two were non-mass lesions and one was not available for histopathologic diagnosis; thus, 137 samples were analyzed.

### 2.3. Histopathological Evaluation

The H&E-stained slides were microscopically evaluated by two pathologists (YU-K and MK-B). Discrepancies were resolved by discussion with a third pathologist (JH-S). Photomicrographs were obtained with a digital camera (Olympus DP22). Histopathological diagnosis was based on *Tumors in domestic animals, 5th edition* [15].

### 2.4. Immunohistochemistry

Immunohistochemistry was performed for CD31 (endothelial cells), CD3 (T cells), PAX5 (B cells), Iba1 (histiocytes), and C-kit (mast cells). Formalin-fixed paraffin-embedded tissues were processed into 4 µm thick sections and picked up with silane-coated slides. The sections were deparaffinized in xylene and rehydrated in graded ethanol before staining. The sections were washed three times in phosphate-buffered saline (PBS) and incubated in 3% hydrogen peroxide prepared in PBS for 20 min at room temperature to block the endogenous hydrogen peroxidase activity. Target antigen retrieval was performed, using a pressure cooker, in a citric acid buffer (pH 6.0) or a Tris-EDTA buffer (pH 9.0) for various periods (Table 1). The sections were cooled in ice water and immersed in 5% normal goat serum for 30 min to block the non-specific binding of antibodies. The sections were then incubated with primary antibodies overnight at 4 °C (Table 1). Then, the sections were washed with PBS and incubated with secondary antibodies (Dako REAL EnVision Kit; Dako, Denmark) for 40 min at room temperature. The labeling was visualized using 3,3’-diaminobenzidine (Dako REAL Envision Kit; Dako, Denmark). The sections were counterstained with Mayer’s hematoxylin, dehydrated in a graded alcohol series, and mounted. Internal positive and negative controls were performed for CD31, CD3, PAX5, and Iba1. A canine cutaneous mast cell tumor was used as an external positive control for C-kit.

## 3. Results

### 3.1. Patient Information

One dog had no age information; the average age of the remaining 136 dogs was 10 years (range: 3–18 years). There were 13 intact males, 13 intact females, 65 castrated males, and 46 spayed females. A total of 127 purebred and 10 mongrel dogs were represented. Most of the purebred dogs were small breeds (n = 112, 82%), and the five breeds that accounted for the highest number were the Maltese (n = 55), toy poodle (n = 17), shih-tzu (n = 10), Pomeranian (n = 8), and Yorkshire terrier (n = 6). Medium to large purebred dogs comprised the cocker spaniel (n = 3), golden retriever (n = 2), Japanese spitz (n = 2), Welsh corgi (n = 2), border collie (n = 1), German shepherd (n = 1), Jindo (n = 1), Labrador retriever (n = 1), standard poodle (n = 1), and whippet (n = 1).

### 3.2. Histopathological Evaluation and Immunohistochemistry

In total, 137 splenic tissue samples were examined. The number of splenic non-neoplastic lesions (n = 99, 72.3%) was more than twice that of neoplastic lesions (n = 38, 27.7%). The histopathological diagnoses of all the samples are summarized in Table 2.

#### 3.2.1. Nodular Hyperplasia

Nodular hyperplasia was diagnosed in 66 dogs in this study. The average age at diagnosis was 9.6 years (range: 4–18 years), and there were 36 males (32 castrated) and 30 females (26 spayed). The main microscopic finding was the proliferation of lymphoid follicles. The nodules were well demarcated and non-infiltrative. The lymphoid follicles were separated from each other and had normal structures, such as the germinal center, mantle zone, and marginal zone (Figure 1A). Most of the lymphocytes that made up the follicles were small, and cellular pleomorphism was minimal. Some cases showed extramedullary hematopoiesis due to the presence of immature erythroid and myeloid cells, including megakaryocytes. The stromal component surrounding the lymphoid follicles comprised less than 50% of the mass.

Immunohistochemistry was performed for CD3 and PAX5, which are markers of T and B cells, respectively. Lymphoid nodular hyperplasia contained T- and B-cell populations in separate areas, correlating with the normal lymphoid follicle anatomy. The T-cell populations were confirmed at the periphery of the follicles (Figure 1B), and B cells were prominently distributed within the follicles (Figure 1C), comprising the germinal centers and the marginal and mantle zones. The distribution of CD3 and PAX5 highlighted the zonation of the follicles.

#### 3.2.2. Hematoma

Hematoma was diagnosed in 33 dogs. The average age at diagnosis was 10 years (range: 3–16 years), and there were 18 males (13 castrated) and 15 females (12 spayed). Histologically, the splenic mass was formed by an accumulation of coagulated blood (Figure 2A). The mass had distinct margins, and stromal cells were dispersed around the mass. Vascular endothelial cells in the mass formed a single layer, and they showed a lack of pleomorphism and mitotic activity (Figure 2B). Hemosiderin-laden macrophages and hematoidin crystals appeared in the mass (Figure 2A). Some cases presented extramedullary hematopoiesis due to the presence of immature erythroid and myeloid cells, including megakaryocytes.

Immunohistochemistry was performed for CD31, a marker of endothelial cells. In splenic hematomas, CD31 was only expressed in a layer of endothelial cells that composed normal blood vessels (Figure 2C).

#### 3.2.3. Splenic Hemangiosarcoma

Fourteen dogs were diagnosed with splenic hemangiosarcoma, and Maltese (n = 8) and toy poodles (n = 2) accounted for most of the cases. The other breeds were the Japanese spitz (n = 1), miniature schnauzer (n = 1), Pomeranian (n = 1), and German shepherd (n = 1). The average age at diagnosis was 11.1 years (range: 8–15 years), and there were eight males (six castrated) and six females (three spayed). In all cases, the tumors were poorly demarcated and infiltrated into the regional parenchyma. Necrosis and bleeding occurred broadly in these lesions. Neoplastic endothelial cells were arranged into irregular vascular spaces and trabecular structures (Figure 3A). Solid areas with no vascular space, formed by the high-density proliferation of tumor cells, were partially observed (Figure 3A). The neoplastic cells were plump and exhibited anisocytosis, anisokaryosis, and prominent nucleoli (Figure 3B).

Immunohistochemistry was performed for CD31. All splenic hemangiosarcoma samples exhibited strong cytoplasmic immunoreactivity with CD31 (Figure 3C). 

#### 3.2.4. Splenic Lymphoma

Splenic lymphoma was diagnosed in 11 dogs, and Maltese (n = 4) and Toy poodles (n = 3) accounted for most of these cases. The other breeds were the bichon frise (n = 1), border collie (n = 1), golden retriever (n = 1), and mongrel (n = 1). The average age at diagnosis was 9.5 years (range: 5–13 years), and there were six males (four castrated) and five females (three spayed). Morphological characteristics in H&E staining were examined and immunohistochemistry for CD3 and PAX5 was performed to classify the splenic lymphoma cases according to the revised European–American lymphoma classification adopted by the World Health Organization. The 11 cases of splenic lymphoma were classified into 6 marginal-zone lymphomas, 1 diffuse large B-cell lymphoma (DLBCL), and 4 peripheral T-cell lymphomas. The marginal-zone lymphomas showed a nodular growth pattern and a characteristic architecture, such as a proliferating marginal zone and fading germinal centers (Figure 4A). The neoplastic cells were mostly composed of intermediate-sized lymphocytes that had nuclei 1.5–2 times the diameter of a red blood cell, with abundant cytoplasm and prominent nucleoli with rare mitotic figures. DLBCL showed a diffuse growth pattern with large lymphocytes that had nuclei more than twice the diameter of a red blood cell. The neoplastic cells exhibited anisokaryosis, prominent nucleoli, and frequent mitotic figures, with eight mitotic figures in an area of one 400× field (0.237 mm^2^) (Figure 4D). Immunohistochemistry in marginal-zone lymphomas and DLBCL confirmed a B-cell origin, given the strong expression of PAX5 and the lack of CD3 immunoreactivity (Figure 4B,C,E,F). The peripheral T-cell lymphomas showed a diffuse growth pattern (Figure 4G), and neoplastic cells were diffusely immunoreactive for CD3 but immunonegative for PAX5 (Figure 4H,I). 

#### 3.2.5. Stromal Sarcomas

Ten dogs were diagnosed with splenic stromal tumors, most of which were Maltese (n = 8). The other breeds were the bichon frise (n = 1) and Labrador retriever (n = 1). The average age at diagnosis was 11.4 years (range: 10–14 years), and there were eight castrated males and two spayed females. Histologically, all stromal sarcomas were poorly demarcated and infiltrated into the surrounding parenchyma. The neoplastic cells were spindle-shaped with collagen deposition and formed broad fascicles that ran in diverse directions (Figure 5A). The cytoplasm was elongated to an oval appearance and showed pleomorphism. The nuclei were variable in size and shape, and some had multiple prominent nucleoli. In one sample from a 14-year-old castrated male Maltese, a distinct feature was identified in some neoplastic cells with clear cytoplasmic fat vacuoles (Figure 5B). This histomorphologic features suggest a diagnosis of liposarcoma, which is encompassed within the broad group of splenic stromal sarcomas. 

Immunohistochemistry was performed for Iba1 and CD31 in all stromal sarcoma samples to differentiate histiocytic sarcoma and hemangiosarcoma. The neoplastic mesenchymal cells showed negative responses to Iba1 and CD31 (Figure 5C,D).

#### 3.2.6. Myelolipoma

Two patients were diagnosed with myelolipoma. The first was an 8-year-old castrated male whippet, and the second was a 13-year-old castrated male miniature schnauzer. Histologically, well-differentiated adipose and hematopoietic tissues were intermixed (Figure 6A). The adipocytes had clear and regular round fat vacuoles and lacked pleomorphism and mitotic activity. Hematopoietic cells comprised immature erythroid and myeloid cells, including megakaryocytes.

#### 3.2.7. Mast Cell Tumor

The spleen of a 15-year-old spayed female mixed-breed dog was diagnosed with a splenic mast cell tumor. The histomorphologic characteristics included multifocal to coalescing nodules of neoplastic mast cells. Marked necrosis and hemorrhages were found throughout the splenic tissue, with distortion of the splenic architecture. The proliferating mast cells were round or polygonal and contained cytoplasmic granules (Figure 6B). In addition, the mast cells showed anisokaryosis and prominent nucleoli. Infiltration of eosinophils and hemosiderin deposition were observed in the lesion. 

Immunohistochemistry was performed for C-kit, a mast cell marker. Neoplastic cells displayed strong cytoplasmic immunoreactivity with C-kit (Figure 6C).

## 4. Discussion

In the present study, the average age of the 136 dogs was 10 years, which suggests a prevalence of splenic mass lesions in older animals. This result was similar to those of other studies in which the mean ages of dogs with splenic mass lesions were 10 to 11 years [13,14,17,18,19,20]. The majority of the splenic samples in this series were collected from small-breed dogs. This likely reflects a prevalence of small-breed dogs in Republic of Korea compared to other geographical locations [21]. It should be noted that there were five cases of splenic tumors of medium to large breed dogs; thus, the results of this study cannot be exclusively applied to small-breed dogs.

Although splenic non-neoplastic lesions occurred more frequently than splenic neoplasia, the incidence of neoplasia was not negligible (28%). The most common splenic mass lesions was nodular hyperplasia (n = 66, 48.2%), with lymphoid nodular hyperplasia being the most common histologic diagnosis, followed by hematoma (n = 33, 24.1%). Consistent with the results of several previous studies [7,9,10,11,12], hemangiosarcoma (n = 14, 10.2%) was the most common splenic neoplasm. However, our cases did not appear to follow the “double two-thirds” rule, which is a commonly quoted descriptor for canine splenic mass lesions [14]. This result suggests that there may be a geographical difference in the distribution of benign and malignant tumors in the spleens of dogs in Republic of Korea.

The incidence of splenic neoplasia in the present study was slightly lower than that reported in an Italian study (30%) [16]. It was previously reported that hemangiosarcoma constitutes the highest proportion of splenic neoplasia (11%), followed by lymphoma (8%) [16], which was confirmed in the present study, as the proportions were 10.2% and 8.0%, respectively. However, the incidence of stromal sarcoma in the present study was 7.3%, which was more than three times higher than the previously reported 2% [16]. The previous study in Italy obtained most of its samples from large breeds (72%). Although most of the samples in our study accounted for a high proportion of small breeds (82%), the present study showed a prevalence of splenic neoplasia similar to the Italian study. However, our cases did not correspond with the findings of another small-breed study [17] that found that 56% of small dogs had splenic neoplasia and that of this subset, 54% were diagnosed with hemangiosarcoma. One study found that a genotype-based breed grouping was a more significant predictor of canine splenic disease than a phenotype-based breed grouping, as applied in our study [14]. Further investigations into more detailed comparisons of splenic mass lesions between small- and large-breed dogs may consider including genotype-based groupings.

Splenic hemangiosarcoma is characterized by aggressive growth and a grave prognosis. A microscopic examination revealed aggressive histomorphologic features in all hemangiosarcoma cases, including poor demarcation of the mass, infiltration, and necrosis. They carry a risk of rupture, which can lead to disseminated metastasis by seeding tumor cells and hemoabdomen. The histologic diagnosis of splenic hemangiosarcoma is usually uncomplicated; however, if identifiable patterns of vessel formation are not apparent (e.g., solid patterns with a lack of vascular spaces), a definitive diagnosis is more difficult. Additionally, the differentiation of splenic hematoma, hemangiosarcoma, and stromal sarcoma is sometimes challenging [15,22]. The application of immunohistochemistry for the definitive identification of endothelial cells (CD31) can help with the diagnosis of hemangiosarcoma. In the present study, all hemangiosarcoma samples exhibited strong cytoplasmic immunoreactivity with CD31. These immunohistochemical features differed significantly from the hematoma cases, ensuring an accurate diagnosis. In some studies, atypical neoplastic endothelial cells were reported to lose immunoreactivity for CD31 [23,24]; however, this was not observed in our cases. In the case of reduced or absent CD31 immunoreactivity in a suspected hemangiosarcoma, additional immunohistochemistry can be performed, including von Willebrand factor, vascular endothelial growth factor A, or angiopoietin-2 [24,25,26]. 

The diagnosis of canine splenic lymphoma is mainly based on histopathological features, but it can be difficult to distinguish some types of lymphoma (e.g., follicular lymphoma, mantle cell lymphoma, and marginal-zone lymphoma) from benign lymphoid hyperplasia. In splenic tissues with ambiguous H&E findings, immunophenotyping with immunohistochemistry for T- and B-cell markers can help differentiate reactive proliferations of lymphocytes from lymphoma. In this study, the differentiation of splenic lymphoma and nodular hyperplasia could be sufficiently performed via immunohistochemical staining with the homogenous expression pattern of either CD3 or PAX5. However, immunohistochemistry for CD3 and PAX5 may not allow for easy distinctions between some cases of lymphoma (especially follicular lymphoma) and nodular hyperplasia. Moreover, the homogenous expression of either CD3 or PAX5 will not apply to null-cell (non-B-cell and non-T-cell) lymphoma. In those cases, immunohistochemistry for Ki-67 and PCR for antigen receptor rearrangements (PARR) can also be useful in differentiating nodular hyperplasia and lymphoma [27,28].

There are many classification systems for canine lymphoid neoplasms, and modifications have been made frequently. Recently, the American College of Veterinary Pathologists Oncology Committee presented a report adapting the World Health Organization system for human lymphoma classification to canine lymphomas [29]. There are various subtypes of T- and B-cell lymphomas, of which marginal-zone lymphoma is the most common in the spleens of dogs [29,30]. Several types of indolent lymphomas, including marginal-zone lymphoma, are associated with long survival after splenectomy [30,31]. Therefore, a precise diagnosis of lymphoma in the canine spleen is of prognostic importance. In the present study, the classification of splenic lymphoma was performed according to this classification, and our findings correlated with the literature in that marginal-zone lymphoma was the most commonly reported subtype. However, because of the small case numbers in our series, these findings should be validated. Additional studies are needed in order to determine additional trends for canine splenic lymphoma in Republic of Korea.

Splenic stromal tumors are a heterogeneous group of mesenchymal neoplasms with a spindle cell morphology. Malignant stromal tumors are more common than benign tumors in dogs [32]. Stromal sarcomas are composed of non-angiomatous and non-lymphoid tumors such as fibrosarcoma, leiomyosarcoma, liposarcoma, myxosarcoma, rhabdomyosarcoma, chondrosarcoma, and osteosarcoma. Stromal sarcomas have a high rate of metastasis and a grave prognosis, with a short MST of 2.5 months [32]. The MSTs of individual sarcoma types were similar for fibrosarcoma (2 months), leiomyosarcoma (3 months), osteosarcoma (1 month), myxosarcoma (2 months), and undifferentiated sarcoma (1 month) [33]. Most splenic stromal tumors can be classified based on histological patterns, but poorly differentiated spindle cell tumors cannot be further categorized. In particular, splenic fibrosarcoma and leiomyosarcoma have many similarities in cell morphology; therefore, it is difficult to distinguish them using histopathology. A histologic examination of stromal sarcomas revealed spindle-shaped neoplastic cells with poorly demarcated masses and infiltration into the surrounding parenchyma. Immunohistochemistry was conducted for Iba1 and CD31 to differentiate histiocytic sarcoma and poorly differentiated hemangiosarcoma, which can have a morphology similar to malignant stromal tumors. All samples of malignant stromal tumors were immunonegative for Iba1 and CD31, ruling out histiocytic sarcoma and hemangiosarcoma, respectively.

There were two cases of myelolipoma. Myelolipoma is a rare tumor in the spleens of dogs. The origin of these masses is debated; while often considered a benign neoplasm, myelolipoma may also represent a metaplastic or hamartomatous process [15].

There was one case of a splenic mast cell tumor. Obtaining a definitive diagnosis based on H&E was difficult because of the presence of additional inflammatory cell types, including hemosiderin-containing macrophages. Immunohistochemistry for C-kit aided the definitive diagnosis by confirming the presence of numerous mast cells within the lesion. The splenic mast cell tumor in our study could not be confirmed to be primary or metastatic, as the history and clinical findings were not provided. The lack of clinical data represents a limitation of this retrospective study.

The distribution of canine splenic mass lesions outlined in the current study will help veterinary clinicians communicate with pet owners regarding prognoses, recommendations for splenectomy, and histopathological diagnoses. In this study, we evaluated canine splenic mass lesions in Republic of Korea, predominantly with samples from small-breed dogs. Since this study also included a small number of medium- and large-breed dogs, the findings here cannot be exclusively applied to small-breed dogs. Additional studies comparing the prevalence of splenic mass lesions between small- and large-breed dogs are needed, including using genotype-based breed groupings (as opposed to phenotype only).

## 5. Conclusions

In the present study, we analyzed the histopathological diagnoses of 137 canine spleens in Republic of Korea for the first time. We also confirmed that using immunohistochemistry aids in obtaining definitive diagnoses of canine splenic tumors. The distribution of non-neoplastic lesions represented approximately 2/3 of the cases, which differed from what has been reported in studies from other geographical regions. These results should be validated within Republic of Korea. The goal of this study was to provide information that will aid veterinary clinicians in their communications with pet owners regarding prognoses and recommendations for splenectomy. Moreover, this study will facilitate additional investigations with more detailed comparisons of splenic mass lesions in small- vs. large-breed dogs or genotype-based groupings of breeds.

## Figures and Tables

**Figure 1 vetsci-10-00247-f001:**
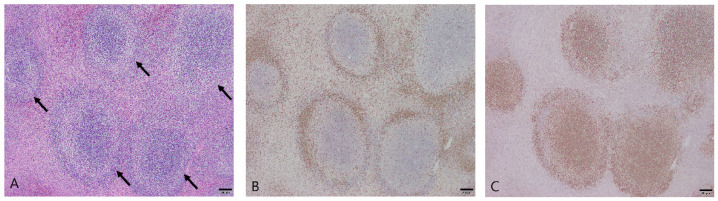
Nodular hyperplasia in the spleen of a dog. (**A**) The proliferating lymphoid follicles were clearly separated from each other (arrows). Hematoxylin and eosin (H&E) at 40× magnification. Scale bar = 200 μm. (**B**) Immunohistochemical staining for CD3 with Mayer’s hematoxylin counterstain. CD3 was expressed at the periphery of the follicles. 40× magnification. Scale bar = 200 μm. (**C**) Immunohistochemical staining for PAX5 with Mayer’s hematoxylin counterstain. PAX5 was expressed within the follicles. 40× magnification. Scale bar = 200 μm.

**Figure 2 vetsci-10-00247-f002:**
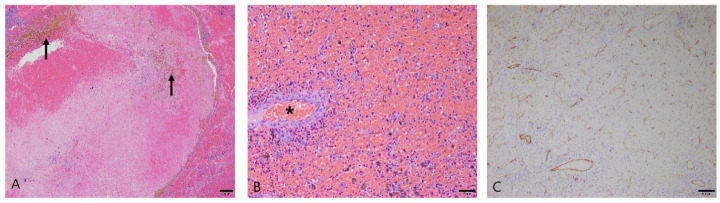
Hematoma in the spleen of dog. (**A**) Splenic mass formed by blood accumulation had yellow hematoidin crystals (arrows). Hematoxylin and eosin (H&E) at 40× magnification. Scale bar = 200 μm. (**B**) The cell population of the mass consisted mostly of red blood cells. Vascular endothelial cells formed a single layer (asterisk). H&E at 200× magnification. Scale bar = 50 μm. (**C**) Immunohistochemical staining for CD31 with Mayer’s hematoxylin counterstain. CD31 expression was only observed in a layer of normal endothelial cells. 100× magnification. Scale bar = 100 μm.

**Figure 3 vetsci-10-00247-f003:**
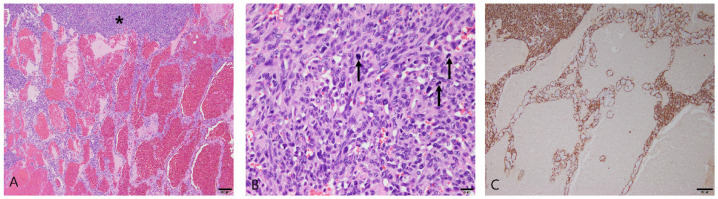
Splenic hemangiosarcoma in the spleen of a dog. (**A**) Irregular vascular and trabecular structures filled the splenic parenchyma. Solid areas with no vascular space were also observed (asterisk). Hematoxylin and eosin (H&E) at 40× magnification. Scale bar = 200 μm. (**B**) The neoplastic endothelial cells were plump, with anaplastic features. Mitotic figures were seen (arrows). H&E at 400× magnification. Scale bar = 20 μm. (**C**) Immunohistochemical staining for CD31 with Mayer’s hematoxylin counterstain. CD31 was strongly expressed in the cell membrane and cytoplasm of neoplastic endothelial cells. 100× magnification. Scale bar = 100 μm.

**Figure 4 vetsci-10-00247-f004:**
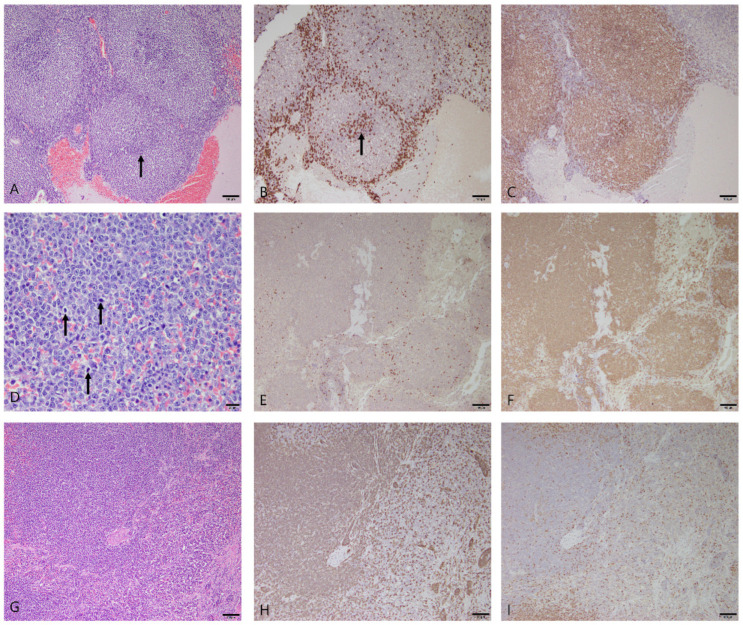
(**A**–**C**) Marginal-zone lymphoma in the spleen of a dog. (**A**) Nodular growth pattern, with proliferating marginal zone and fading germinal center (arrow). Hematoxylin and eosin (H&E) at 100× magnification. Scale bar = 100 μm. (**B**) Immunohistochemical staining for CD3 with Mayer’s hematoxylin counterstain. CD3-immunopositive cells were scattered at the periphery of the follicles, with small clusters in the fading germinal center (arrow). 100× magnification. Scale bar = 100 μm. (**C**) Immunohistochemical staining for PAX5 with Mayer’s hematoxylin counterstain. PAX5 was prominently expressed in the follicles. 100× magnification. Scale bar = 100 μm. (**D**–**F**) Diffuse large B-cell lymphoma in the spleen of a dog. (**D**) Large lymphocytes showed severe anisokaryosis, with prominent nucleoli and frequent mitotic figures (arrows). H&E at 400× magnification. Scale bar = 20 μm. (**E**) Immunohistochemical staining for CD3 with Mayer’s hematoxylin counterstain. Neoplastic cells were immunonegative for CD3. 100× magnification. Scale bar = 100 μm. (**F**) Immunohistochemical staining for PAX5 with Mayer’s hematoxylin counterstain. Neoplastic cells were diffusely immunopositive for PAX5. 100× magnification. Scale bar = 100 μm. (**G**–**I**) Peripheral T-cell lymphoma in the spleen a dog. (**G**) Diffuse growth pattern of neoplastic lymphocytes was observed. H&E at 100× magnification. Scale bar = 100 μm. (**H**) Immunohistochemical staining for CD3 with Mayer’s hematoxylin counterstain. Neoplastic cells were diffusely immunopositive for CD3. 100× magnification. Scale bar = 100 μm. (**I**) Immunohistochemical staining for PAX5 with Mayer’s hematoxylin counterstain. Neoplastic cells were immunonegative for PAX5. 100× magnification. Scale bar = 100 μm.

**Figure 5 vetsci-10-00247-f005:**
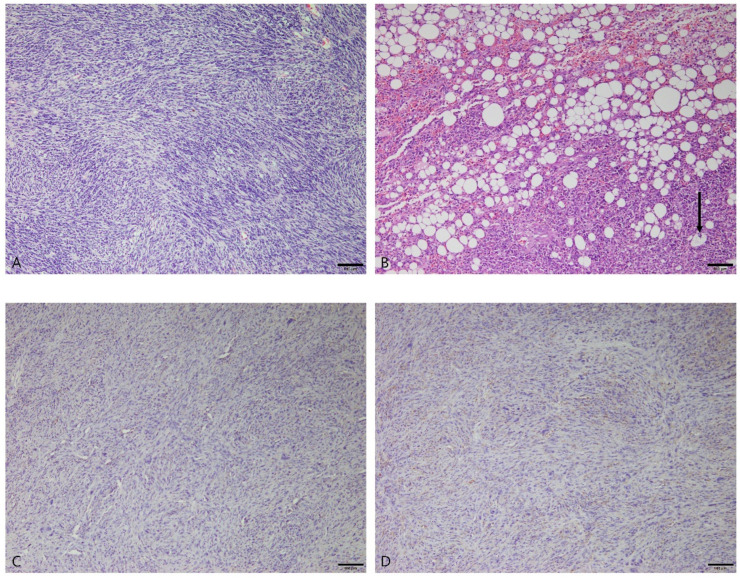
Stromal sarcomas in the spleens of different dogs. (**A**) The spindle-shaped neoplastic cells formed haphazard, interlacing fascicles. Hematoxylin and eosin (H&E) at 100× magnification. Scale bar = 100 μm. (**B**) The mesenchymal neoplastic cells contained clear cytoplasmic fat vacuoles. Bizarre nuclei were observed within lipocytes (arrow). H&E at 100× magnification. Scale bar = 100 μm. (**C**) Immunohistochemical staining for Iba1 with Mayer’s hematoxylin counterstain. Neoplastic cells were immunonegative for Iba1. 100× magnification. Scale bar = 100 μm. (**D**) Immunohistochemical staining for CD31 with Mayer’s hematoxylin counterstain. Neoplastic cells were immunonegative for CD31. 100× magnification. Scale bar = 100 μm.

**Figure 6 vetsci-10-00247-f006:**
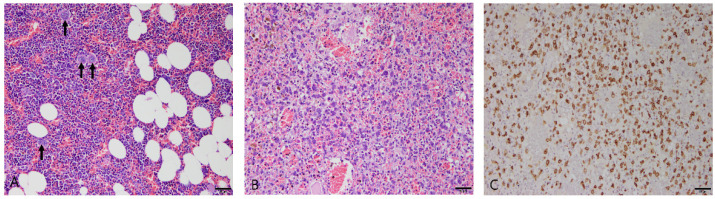
(**A**) Myelolipoma in the spleen of a dog. Immature hematopoietic cells were intermixed with well-differentiated adipocytes. Megakaryocytes were observed (arrows). Hematoxylin and eosin (H&E) at 200× magnification. Scale bar = 50 μm. (**B**,**C**) Splenic mast cell tumor in the spleen of a dog. (**B**) Mast cells with basophilic cytoplasmic granules proliferated in the splenic parenchyma. H&E at 200× magnification. Scale bar = 50 μm. (**C**) Immunohistochemistry staining for C-kit with Mayer’s hematoxylin counterstain. The neoplastic mast cells showed strong cytoplasmic immunoreactivity with C-kit. 200× magnification. Scale bar = 50 μm.

**Table 1 vetsci-10-00247-t001:** Primary antibodies and immunohistochemical staining protocols.

Antibody	Supplier	Clone	Isotype	Dilution	Antigen Retrieval
CD31	Dako	JC70A	Mouse IgG1	1:100	Tris-EDTA (15 min)
CD3	Dako	Polyclonal	Rabbit polyclonal	1:200	Citric acid (15 min)
PAX5	BD Biosciences	24/Pax-5	Mouse IgG1	1:100	Citric acid (15 min)
Iba1	Fujifilm	Polyclonal	Rabbit polyclonal	1:2000	Citric acid (15 min)
C-kit	Dako	Polyclonal	Rabbit polyclonal	1:400	Citric acid (15 min)

**Table 2 vetsci-10-00247-t002:** Histopathological diagnoses of 137 canine splenic tissues.

Diagnosis		Number	Percentage	Total
Non-neoplastic	Nodular hyperplasia	66	48.2%	99 (72.3%)
	Hematoma	33	24.1%	
Neoplastic	Hemangiosarcoma	14	10.2%	38 (27.7%)
	Lymphoma	11	8.0%	
	Stromal sarcomas	10	7.3%	
	Myelolipoma	2	1.5%	
	Mast cell tumor	1	0.7%	
Total		137		

## Data Availability

The data presented in this study are available on request from the corresponding author.

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
