# Peer review of "Analysis of the Prevalence of Canine Splenic Mass Lesions in Republic of Korea via Histopathological Diagnosis with Immunohistochemistry"

_vetsci, 2023, doi:10.3390/vetsci10040247_

Round 1

Reviewer 1 Report

Introduction omits studies on small breed dogs, such as: SPLENIC MASS DIAGNOSIS IN DOGS UNDERGOING SPLENECTOMY ACCORDING TO BREED SIZE. O'Byrne Kadie, Hosgood Giselle, Vet Rec. 2019 May 18; 184(20):620, and OUTCOMES OF 43 SMALL BREED DOGS TREATED FOR SPLENIC HEMANGIOSARCOMA. Story AL, Wavreille V, Abrams B, et al, Vet Surg. 2020 Aug;49(6):1154-1163. Other studies do exist.

Sample collection: Samples were collected randomly in a time period between Jan and Oct 2022. Does this mean you picked randomly all samples between Jan and Oct 2022 or you picked 140 samples from all (how many then?) samples between Jan and Oct 2022, and then you had by coincidence mainly small breed dogs? Why did you not decide to analyze only small breed dogs, if that is representative in Korea?

Classically results are presented withouth presentation/description of scientific knowledge, and without citing pertinent literature. This may be part of the introduction or discussion, or it can be assumed to be known by the reader.

line 330 change Italy to an Italian study, as the cited study does not represent the whole Italian dog population.

Conclusion: How will the result of this study help veterinarians with the treatment decision? Prior to histopathological analysis the nature of a splenic disease is unknown except cytology was performed, and would give a hint. Cytology would not be applicable in the presence of a hemoabdomen due to splenic ruptured mass.

I do not see the primary help for treatment decisions, as ultimately histopath is the goldstandard for splenic disease. It may probably help with prognosis, as 72% to 28% was non-neoplastic vs neoplastic. Ultimately these are always speculations, and only histopathology will reveal which dog is a lucky one.

I would be happy with another presentation of the results. 

Author Response

We revised our manuscript according to your comments. Please see the attachment.

We really appreciate your detailed correction.

Reviewer 2 Report

In the present manuscript, the authors make, in small canine breeds, the histopathological diagnosis of splenic diseases and the immunohistochemical evaluation for specific tumor markers for a more accurate differential diagnosis. The aim is that the results can help veterinarians in the treatment of canine spleen diseases and for the comparison of splenic tumors between small- and large-breed dogs.

The work is very well written and presented, in a clear and concise way, allowing its replication. In the Results section, it seems very good to me at the beginning of each pathology, an introductory paragraph about it, which helps to frame and inform any reader. Photomicrographs are of very acceptable quality.

I have just a few suggestions to make that I think might improve the manuscript:

Comment 1. In the IHC section refer to the controls used.

Comment 2. Refer how the slides were observed and photomicrographs obtained (microscope and digital camera?).

Comment 3: The legends of Figs 1, 2, 3, 7 and 8 would help the reader if some of the finds were identified with some notation e.g. arrows, asterisks, arrow heads (as in Figs. 4, 5 and 6).

Comment 4. In the legend to Figure 9-12, please, write [Mayer's hematoxylin counterstain]

Comment 5. In the Conclusions, it would have been interesting to define an index, based on the expression of the markers used, that could help the veterinarian in a more direct way for the therapy to be carried out, including, or not, splenectomy.

Author Response

(The authors gave the same response as above.)

Reviewer 3 Report

The authors have analyzed a retrospective series of 139 canine spleens using conventional histopathology and immunohistochemistry (IHC) with 5 differentiation markers, in order to determine the relative frequency of canine splenic disorders in their country, South Korea. The breed distribution of Korean dogs markedly differs from the one of Western countries, with overrepresentation of toy and small breeds, and underrepresentation of medium and large breeds, including German Shepherd dogs, Labrador Retrievers, Golden Retrievers, Dobermanns, Boxers, and Bernese Mountain dogs. The results indicate that non-neoplastic changes (n=101, 73%) predominated over splenic tumors (n=38, 27%). The 5 most common conditions were splenic nodular hyperplasia, hematoma, hemangiosarcoma, lymphoma, and stromal sarcoma. 

This study is interesting because good case series are needed from all over the world to gain insight into geographical disparities and possible new breed predispositions to some lesions. This manuscript is very well written and illustrated. In my opinion, this manuscript deserves publication in Veterinary Sciences, although I have some reservations about the incomplete characterization of splenic lymphomas and stromal sarcomas.

Minor comments

1.         Page 2, lines 52-53, introduction: maybe add “myeloid” in this sentence, to account for possible diagnoses of splenic mast cell tumors, histiocytic sarcomas, and myeloid leukemias.

2.         Page 3, line 93, methods: either in this sentence or in table 1, please indicate that these IHC markers are specific for endothelial cells, T cells, B cells, connective tissue cells and mast cells. 

3.         Page 4, lines 152-153, hematoma: I agree that hematomas are composed of coagulated blood, i.e., large amounts of red blood cells and fibrin, and sparse entrapped leucocytes. Line 153, “fibrin deposition” is thus a redundancy with “coagulated blood”. When necrosis was present, is it sure that the diagnosis was hematoma, or could it be a splenic infarct? 

4.         Page 4, lines 162-163, splenitis: there is some redundancy between “purulent”, “neutrophilic”, and “fibrino-suppurative”. All 3 could be gathered under the denomination “fibrinous/suppurative”. What about acute necrotic / hemorrhagic splenitis? 

5.         Page 5, figures 1–3: please write in the legend the lengths of scale bars, as this is not readable on the pictures. 

6.         Page 5, figures 1–3: the disposition of figure panels is very difficult to apprehend by readers. I would suggest that the 6 panels are renamed Fig 1 A–F (non-neoplastic splenic lesions), and that fig 1A (nodular hyperplasia, low magnification) and fig 1B (nodular hyperplasia, high magnification) are put on a same line, on the left (1A) and right (1B) of the page. Same comment for figures 4–6 page 6.

7.         Pages 4-5, lines 168-170, splenitis: were infectious agents also searched for using Gram stain, Fite stain, and/or Grocott-Gomori methenamine silver stain? 

8.         Page 5, line 199, hemangiosarcomas: it is also possible to add here the histological grade of the 14 diagnosed splenic hemangiosarcomas (Ogilvie GK, Powers BE, Mallinckrodt CH, Withrow SJ. Surgery and doxorubicin in dogs with hemangiosarcoma. J Vet Intern Med. 1996;10(6):379-84. doi: 10.1111/j.1939-1676.1996.tb02085.x.)

9.         Page 5, line 203,and page 6, line 205, lymphoma: only follicular lymphomas, and sometimes mantle cell lymphomas and marginal zone lymphomas, can be difficult to distinguish from lymphoid hyperplasia, not all lymphomas. For the differential diagnosis between lymphoid hyperplasia and follicular lymphoma, Ki-67 immunohistochemistry is very useful too (reference in human oncopathology: Bryant RJ, Banks PM, O'Malley DP. Ki67 staining pattern as a diagnostic tool in the evaluation of lymphoproliferative disorders. Histopathology. 2006;48(5):505-15. doi: 10.1111/j.1365-2559.2006.02378.x.) 

10.      Page 6, lines 212-215, lymphomas, and page 7, lines 240-244, legend for figure 5: I do not agree with the histological description of lymphomas, because it is impossible to describe “lymphoma” as a whole. Here, it is important to specify which types of lymphomas have been diagnosed (at least for the most common types, indolent B-cell: follicular, mantle cell, marginal zone; aggressive B-cell: diffuse large B-cell, and T-cell lymphomas). The authors can also choose to present how many cases were nodular versus diffuse, small-cell versus medium-cell versus large-cell, and with low / intermediate / high mitotic counts.

11.      Page 7, lines 263-269, mast cell tumor: please indicate if this MCT was a splenic metastasis of a cutaneous or gastro-intestinal MCT, or could correspond to a mast cell leukemia with splenic involvement. In this paragraph, please cite “Figure 8”. 

12.      Pages 7-8, paragraph 3.3 IHC, please consider adding IHC results in each lesion description (lymphoid hyperplasia, hematoma, hemangiosarcoma, stromal sarcomas and mast cell tumors), and delete section 3.3.

13.      Page 8, Figure 10C–D: there seems to be an inversion between fig 10D (Pax5 in a B-cell lymphoma) and fig 10C (CD3 in a B-cell lymphoma).

14.      Page 8, lines 284-288: I do not fully agree with the IHC description of lymphoid hyperplasia. T- and B-cells in lymphoid hyperplasia are actually intermingled, not “clearly separated”, although they are located in different areas. I have trouble with “marginal T-cell zone”, because the marginal zone belongs to lymphoid follicles, thus to B-cell zones (even if it contains T cells). I agree with the authors that CD3/Pax5 highlights the zonation of the white pulp, which is maintained in nodular hyperplasia, with B-cells very abundant in lymphoid follicles (germinal centers + mantle zone + marginal zone), and T cells abundant outside follicles. Furthermore, the IHC provided would not allow for easy distinction between a follicular lymphoma, and an atypical lymphoid hyperplasia with partial loss/fading of mantle zones. I personally use Ki-67 in this purpose, as Bcl-2 is not reliable in this indication (lack of Bcl-2 overexpression in canine and feline follicular lymphomas). 

15.      Page 8, lines 293-296, vimentin IHC: note that vimentin IHC is not informative in spindle cell sarcomas: strong vimentin expression is expected, but does not give any further diagnostic or prognostic indication. Please consider submitting the 11 stromal sarcomas to additional markers such as smooth muscle actin for leiomyosarcomas, S100 protein for malignant nerve sheath tumors, and Iba-1 or CD204 for histiocytic sarcomas (which can be spindle-cell sarcomas on a morphological basis). 

16.      Results: was there a difference in patient age between the 101 dogs with non-neoplastic splenic lesions and the 38 dogs with splenic tumors?

17.      Page 10, lines 353-355, Discussion: I do not agree at all with the sentence “splenic lymphoma is associated with the proliferation of lymphoid follicles”! Most B-cell lymphomas are not follicular, and by definition T-cell lymphomas do not originate from follicular cells. In most lymphomas, lymphoid follicles do not loose their polarity and merge with each other, but are compressed, fade, and disappear; the sentence line 354 mostly applies to follicular lymphomas, which are rare in veterinary oncology. Lines 354-355: large-cell lymphomas are not supposed to show anaplastic characteristics, unless they are anaplastic large-cell lymphomas: they can be centroblastic or immunoblastic for instance. The authors can also discuss the fact that “homogenous expression of either CD3 or Pax5” (line 360) does not apply to T-cell rich B-cell lymphomas; the existence of nul (non-B non-T) lymphomas; and the possible use of PARR in case of doubt between lymphoid hyperplasia and lymphoma in dogs. Finally, to emphasize the importance of a precise diagnosis of lymphoma in canine spleens, it can be pointed out that the spleen hosts several types of indolent lymphomas associated with long survival after splenectomy (Valli VE, Vernau W, de Lorimier LP, Graham PS, Moore PF. Canine indolent nodular lymphoma. Vet Pathol. 2006;43(3):241-56. doi: 10.1354/vp.43-3-241. PMID: 16672571.)

18.      Discussion: can the authors please discuss the absence / underrepresentation of some splenic lesions such as splenic infarcts / thrombosis, and histiocytic sarcoma? Splenitis was not discussed at all; does leishmaniasis exists in Korea? The authors emphasize on small breed overrepresentation in Korea compared to other geographical areas, but breed predispositions to the observed splenic lesions (especially tumors) are not dicussed.

Typos / Spelling

-       Page 1, line 42, introduction: “histological processing” may be more appropriate than “histopathological examination” here.

-       Page 3, last line, in Table 1: stromal sarcomas, “7.2%” instead of “6.5%).

-       Page 4, line 137: “of the mass” may not be necessary here.

-       Page 5, line 172, legend for figure 1: maybe “proliferating” instead of “proliferated”. 

-       Page 5, lines 177-181, legend for figure 3: maybe the legend for Fig 3A could be “neutrophilic splenitis”, and the legend for Fig 3B “granulomatous splenitis”, to make it shorter. 

-       Page 5, line 195: maybe “were arranged into” instead of “showed”. Line 197: the dot can be deleted after “observed”. 

-       Page 7, line 260, myelipomas: suggestion “immature or mature erythroid and myeloid cells, including megakaryocytes” (because megakaryocytes are myeloid cells).

-       Page 12, line 417, reference 1: the end of the reference is Exp Lung Res. 1981;2(3):231-8. doi: 10.3109/01902148109052318.

-       Page 12, line 443, reference 13: the end of the reference is Europ J Vet Pathol. 2001;7(3):101-109. 

-       Page 12, line 449, reference 16: the journal name is Vet Pathol and the year of publication is 2017.

-       Page 12, line 456, reference 19: the end of the reference is J Vet Med Sci. 2018;80(2):213-218. doi: 10.1292/jvms.17-0561. 

-       Page 12, line 462, reference 21: the correct doi is doi: 10.1177/0300985810379428.

Author Response

(The authors gave the same response as above.)
